# On Variance Reduction in Stochastic Gradient Descent and its Asynchronous Variants

**Sashank J. Reddi**
Carnegie Mellon University
sjakkamr@cs.cmu.edu

**Ahmed Hefny**
Carnegie Mellon University
ahefny@cs.cmu.edu

**Suvrit Sra**
Massachusetts Institute of Technology
suvrit@mit.edu

**Barnabás Póczos**
Carnegie Mellon University
bapoczos@cs.cmu.edu

**Alex Smola**
Carnegie Mellon University
alex@smola.org

## Abstract

We study optimization algorithms based on variance reduction for stochastic gradient descent (SGD). Remarkable recent progress has been made in this direction through development of algorithms like SAG, SVRG, SAGA. These algorithms have been shown to outperform SGD, both theoretically and empirically. However, asynchronous versions of these algorithms—a crucial requirement for modern large-scale applications—have not been studied. We bridge this gap by presenting a unifying framework for many variance reduction techniques. Subsequently, we propose an asynchronous algorithm grounded in our framework, and prove its fast convergence. An important consequence of our general approach is that it yields asynchronous versions of variance reduction algorithms such as SVRG and SAGA as a byproduct. Our method achieves near linear speedup in sparse settings common to machine learning. We demonstrate the empirical performance of our method through a concrete realization of asynchronous SVRG.

## 1 Introduction

There has been a steep rise in recent work [6, 7, 9–12, 25, 27, 29] on "variance reduced" stochastic gradient algorithms for convex problems of the *finite-sum* form:

$$\min_{x \in \mathbb{R}^d} \ f(x) := \tfrac{1}{n} \sum_{i=1}^{n} f_i(x). \tag{1.1}$$

Under strong convexity assumptions, such variance reduced (VR) stochastic algorithms attain better convergence rates (in expectation) than stochastic gradient descent (SGD) [18, 24], both in theory and practice.[1] The key property of these VR algorithms is that by exploiting problem structure and by making suitable space-time tradeoffs, they reduce the variance incurred due to stochastic gradients. This variance reduction has powerful consequences: it helps VR stochastic methods attain linear convergence rates, and thereby circumvents slowdowns that usually hit SGD.

Although these advances have great value in general, for large-scale problems we still require parallel or distributed processing. And in this setting, asynchronous variants of SGD remain indispensable [2, 8, 13, 21, 28, 30]. Therefore, a key question is how to extend the synchronous finite-sum VR algorithms to asynchronous parallel and distributed settings.

We answer one part of this question by developing new asynchronous parallel stochastic gradient methods that provably converge at a linear rate for smooth strongly convex finite-sum problems. Our methods are inspired by the influential SVRG [10], S2GD [12], SAG [25] and SAGA [6] family of algorithms. We list our contributions more precisely below.

**Contributions.** Our paper makes two core contributions: (i) a formal general framework for variance reduced stochastic methods based on discussions in [6]; and (ii) asynchronous parallel VR algorithms within this framework. Our general framework presents a formal unifying view of several VR methods (e.g., it includes SAGA and SVRG as special cases) while expressing key algorithmic and practical tradeoffs concisely. Thus, it yields a broader understanding of VR methods, which helps us obtain *asynchronous parallel* variants of VR methods. Under sparse-data settings common to machine learning problems, our parallel algorithms attain speedups that scale near linearly with the number of processors.

As a concrete illustration, we present a specialization to an asynchronous SVRG-like method. We compare this specialization with non-variance reduced asynchronous SGD methods, and observe strong empirical speedups that agree with the theory.

**Related work.** As already mentioned, our work is closest to (and generalizes) SAG [25], SAGA [6], SVRG [10] and S2GD [12], which are primal methods. Also closely related are dual methods such as SDCA [27] and Finito [7], and in its convex incarnation MISO [16]; a more precise relation between these dual methods and VR stochastic methods is described in Defazio's thesis [5]. By their algorithmic structure, these VR methods trace back to classical non-stochastic incremental gradient algorithms [4], but by now it is well-recognized that randomization helps obtain much sharper convergence results (in expectation). Proximal [29] and accelerated VR methods have also been proposed [20, 26]; we leave a study of such variants of our framework as future work. Finally, there is recent work on lower-bounds for finite-sum problems [1].

Within asynchronous SGD algorithms, both parallel [21] and distributed [2, 17] variants are known. In this paper, we focus our attention on the parallel setting. A different line of methods is that of (primal) coordinate descent methods, and their parallel and distributed variants [14, 15, 19, 22, 23]. Our asynchronous methods share some structural assumptions with these methods. Finally, the recent work [11] generalizes S2GD to the mini-batch setting, thereby also permitting parallel processing, albeit with more synchronization and allowing only small mini-batches.

## 2 A General Framework for VR Stochastic Methods

We focus on instances of (1.1) where the cost function $f(x)$ has an $L$-Lipschitz gradient, so that $\|\nabla f(x) - \nabla f(y)\| \le L\|x - y\|$, and it is $\lambda$-strongly convex, i.e., for all $x, y \in \mathbb{R}^d$,

$$f(x) \ge f(y) + \langle \nabla f(y), x - y \rangle + \frac{\lambda}{2}\|x - y\|^2. \tag{2.1}$$

While our analysis focuses on strongly convex functions, we can extend it to just smooth convex functions along the lines of [6, 29].

Inspired by the discussion on a general view of variance reduced techniques in [6], we now describe a formal general framework for variance reduction in stochastic gradient descent. We denote the collection $\{f_i\}_{i=1}^n$ of functions that make up $f$ in (1.1) by $\mathcal{F}$. For our algorithm, we maintain an additional parameter $\alpha_i^t \in \mathbb{R}^d$ for each $f_i \in \mathcal{F}$. We use $A^t$ to denote $\{\alpha_i^t\}_{i=1}^n$. The general iterative framework for updating the parameters is presented as Algorithm 1. Observe that the algorithm is still abstract, since it does not specify the subroutine SCHEDULEUPDATE. This subroutine determines the crucial update mechanism of $\{\alpha_i^t\}$ (and thereby of $A^t$). As we will see different schedules give rise to different fast first-order methods proposed in the literature. The part of the update based on $A^t$ is the key for these approaches and is responsible for variance reduction.

Next, we provide different instantiations of the framework and construct a new algorithm derived from it. In particular, we consider incremental methods SAG [25], SVRG [10] and SAGA [6], and classic gradient descent GRADIENTDESCENT for demonstrating our framework.

**ALGORITHM 1:** GENERIC STOCHASTIC VARIANCE REDUCTION ALGORITHM

---

**Data**: $x^0 \in \mathbb{R}^d, \alpha_i^0 = x^0 \;\; \forall i \in [n] \triangleq \{1, \ldots, n\}$, step size $\eta > 0$
Randomly pick a $I_T = \{i_0, \ldots, i_T\}$ where $i_t \in \{1, \ldots, n\} \; \forall \, t \in \{0, \ldots, T\}$ ;
**for** $t = 0$ **to** $T$ **do**
    Update iterate as $x^{t+1} \leftarrow x^t - \eta \left( \nabla f_{i_t}(x^t) - \nabla f_{i_t}(\alpha_{i_t}^t) + \frac{1}{n} \sum_i \nabla f_i(\alpha_i^t) \right)$ ;
    $A^{t+1} = \text{SCHEDULEUPDATE}(\{x^i\}_{i=0}^{t+1}, A^t, t, I_T)$ ;
**end**

**return** $x^T$

---

Figure 1 shows the schedules for the aforementioned algorithms. In case of SVRG, SCHEDULEUPDATE is triggered every $m$ iterations (here $m$ denotes precisely the number of inner iterations used in [10]); so $A^t$ remains unchanged for the $m$ iterations and all $\alpha_i^t$ are updated to the current iterate at the $m^{\text{th}}$ iteration. For SAGA, unlike SVRG, $A^t$ changes at the $t^{th}$ iteration for all $t \in [T]$. This change is only to a single element of $A^t$, and is determined by the index $i_t$ (the function chosen at iteration $t$). The update of SAG is similar to SAGA insofar that only one of the $\alpha_i$ is updated at each iteration. However, the update for $A^{t+1}$ is based on $i_{t+1}$ rather than $i_t$. This results in a biased estimate of the gradient, unlike SVRG and SAGA. Finally, the schedule for gradient descent is similar to SAG, except that all the $\alpha_i$'s are updated at each iteration. Due to the full update we end up with the exact gradient at each iteration. This discussion highlights how the scheduler determines the resulting gradient method.

To motivate the design of another schedule, let us consider the computational and storage costs of each of these algorithms. For SVRG, since we update $A^t$ after every $m$ iterations, it is enough to store a full gradient, and hence, the storage cost is $O(d)$. However, the running time is $O(d)$ at each iteration and $O(nd)$ at the end of each epoch (for calculating the full gradient at the end of each epoch). In contrast, both SAG and SAGA have high storage costs of $O(nd)$ and running time of $O(d)$ per iteration. Finally, GRADIENTDESCENT has low storage cost since it needs to store the gradient at $O(d)$ cost, but very high computational costs of $O(nd)$ at *each* iteration.

SVRG has an additional computation overhead at the end of each epoch due to calculation of the whole gradient. This is avoided in SAG and SAGA at the cost of additional storage. When $m$ is very large, the additional computational overhead of SVRG amortized over all the iterations is small. However, as we will later see, this comes at the expense of slower convergence to the optimal solution. The tradeoffs between the epoch size $m$, additional storage, frequency of updates, and the convergence to the optimal solution are still not completely resolved.

---

**SVRG:**SCHEDULEUPDATE$(\{x^i\}_{i=0}^{t+1}, A^t, t, I_T)$
**for** *i = 1 to n* **do**
    $\alpha_i^{t+1} = \mathbb{1}(m \mid t)x^t + \mathbb{1}(m \nmid t)\alpha_i^t$ ;
**end**
**return** $A^{t+1}$

**SAGA:**SCHEDULEUPDATE$(\{x^i\}_{i=0}^{t+1}, A^t, t, I_T)$
**for** *i = 1 to n* **do**
    $\alpha_i^{t+1} = \mathbb{1}(i_t = i)x^t + \mathbb{1}(i_t \neq i)\alpha_i^t$ ;
**end**
**return** $A^{t+1}$

**SAG:**SCHEDULEUPDATE$(\{x^i\}_{i=0}^{t+1}, A^t, t, I_T)$
**for** *i = 1 to n* **do**
    $\alpha_i^{t+1} = \mathbb{1}(i_{t+1} = i)x^{t+1} + \mathbb{1}(i_{t+1} \neq i)\alpha_i^t$ ;
**end**
**return** $A^{t+1}$

**GD:**SCHEDULEUPDATE$(\{x^i\}_{i=0}^{t+1}, A^t, t, I_T)$
**for** *i = 1 to n* **do**
    $\alpha_i^{t+1} = x^{t+1}$ ;
**end**
**return** $A^{t+1}$

Figure 1: SCHEDULEUPDATE function for SVRG (top left), SAGA (top right), SAG (bottom left) and GRADIENTDESCENT (bottom right). While SVRG is epoch-based, rest of algorithms perform updates at each iteration. Here $a|b$ denotes that $a$ divides $b$.

---

A straightforward approach to design a new scheduler is to combine the schedules of the above algorithms. This allows us to tradeoff between the various aforementioned parameters of our interest. We call this schedule *hybrid stochastic average gradient* (HSAG). Here, we use the schedules of SVRG and SAGA to develop HSAG. However, in general, schedules of any of these algorithms can

$$\boxed{\begin{array}{l} \textbf{HSAG:}\textsc{ScheduleUpdate}(x^t, A^t, t, I_T) \\ \textbf{for } i = 1 \text{ to } n \textbf{ do} \\ \quad \left| \quad \alpha_i^{t+1} = \left\{ \begin{array}{ll} \mathbb{1}(i_t = i)x^t + \mathbb{1}(i_t \neq i)\alpha_i^t & \text{if } i \in S \\ \mathbb{1}(s_i \mid t)x^t + \mathbb{1}(s_i \nmid t)\alpha_i^t & \text{if } i \notin S \end{array} \right. \right. \\ \textbf{end} \\ \textbf{return } A^{t+1} \end{array}}$$

Figure 2: SCHEDULEUPDATE for HSAG. This algorithm assumes access to some index set $S$ and the schedule frequency vector $s$. Recall that $a|b$ denotes $a$ divides $b$

be combined to obtain a hybrid algorithm. Consider some $S \subseteq [n]$, the indices that follow SAGA schedule. We assume that the rest of the indices follow an SVRG-like schedule with *schedule frequency* $s_i$ for all $i \in \overline{S} \triangleq [n] \setminus S$. Figure 2 shows the corresponding update schedule of HSAG. If $S = [n]$ then HSAG is equivalent to SAGA, while at the other extreme, for $S = \emptyset$ and $s_i = m$ for all $i \in [n]$, it corresponds to SVRG. HSAG exhibits interesting storage, computational and convergence trade-offs that depend on $S$. In general, while large cardinality of $S$ likely incurs high storage costs, the computational cost per iteration is relatively low. On the other hand, when cardinality of $S$ is small and $s_i$'s are large, storage costs are low but the convergence typically slows down.

Before concluding our discussion on the general framework, we would like to draw the reader's attention to the advantages of studying Algorithm 1. First, note that Algorithm 1 provides a unifying framework for many incremental/stochastic gradient methods proposed in the literature. Second, and more importantly, it provides a generic platform for analyzing this class of algorithms. As we will see in Section 3, this helps us develop and analyze asynchronous versions for different finite-sum algorithms under a common umbrella. Finally, it provides a mechanism to derive new algorithms by designing more sophisticated schedules; as noted above, one such construction gives rise to HSAG.

## 2.1 Convergence Analysis

In this section, we provide convergence analysis for Algorithm 1 with HSAG schedules. As observed earlier, SVRG and SAGA are special cases of this setup. Our analysis assumes unbiasedness of the gradient estimates at each iteration, so it does not encompass SAG. For ease of exposition, we assume that all $s_i = m$ for all $i \in [n]$. Since HSAG is epoch-based, our analysis focuses on the iterates obtained after each epoch. Similar to [10] (see Option II of SVRG in [10]), our analysis will be for the case where the iterate at the end of $(k+1)^{\text{st}}$ epoch, $x^{km+m}$, is replaced with an element chosen randomly from $\{x^{km}, \dots, x^{km+m-1}\}$ with probability $\{p_1, \dots, p_m\}$. For brevity, we use $\tilde{x}^k$ to denote the iterate chosen at the $k^{\text{th}}$ *epoch*. We also need the following quantity for our analysis:

$$\tilde{G}_k \triangleq \frac{1}{n} \sum_{i \in S} \left( f_i(\alpha_i^{km}) - f_i(x^*) - \langle \nabla f_i(x^*), \alpha_i^{km} - x^* \rangle \right).$$

**Theorem 1.** *For any positive parameters* $c, \beta, \kappa > 1$, *step size* $\eta$ *and epoch size* $m$, *we define the following quantities:*

$$\gamma = \kappa \left[ 1 - \left( 1 - \frac{1}{\kappa} \right)^m \right] \left( 2c\eta(1 - L\eta(1+\beta)) - \frac{1}{n} - \frac{2c}{\kappa\lambda} \right)$$

$$\theta = \max \left\{ \left[ \frac{2c}{\gamma\lambda} \left( 1 - \frac{1}{\kappa} \right)^m + \frac{2Lc\eta^2}{\gamma} \left( 1 + \frac{1}{\beta} \right) \kappa \left[ 1 - \left( 1 - \frac{1}{\kappa} \right)^m \right] \right], \left( 1 - \frac{1}{\kappa} \right)^m \right\}.$$

*Suppose the probabilities* $p_i \propto (1 - \frac{1}{\kappa})^{m-i}$, *and that* $c, \beta, \kappa$, *step size* $\eta$ *and epoch size* $m$ *are chosen such that the following conditions are satisfied:*

$$\frac{1}{\kappa} + 2Lc\eta^2 \left( 1 + \frac{1}{\beta} \right) \leq \frac{1}{n}, \ \gamma > 0, \ \theta < 1.$$

*Then, for iterates of Algorithm 1 under the* HSAG *schedule, we have*

$$\mathbb{E} \left[ f(\tilde{x}^{k+1}) - f(x^*) + \frac{1}{\gamma} \tilde{G}_{k+1} \right] \leq \theta \, \mathbb{E} \left[ f(\tilde{x}^k) - f(x^*) + \frac{1}{\gamma} \tilde{G}_k \right].$$

As a corollary, we immediately obtain an expected linear rate of convergence for HSAG.

**Corollary 1.** *Note that $\tilde{G}_k \geq 0$ and therefore, under the conditions specified in Theorem 1 and with $\bar{\theta} = \theta (1 + 1/\gamma) < 1$ we have*

$$\mathbb{E}\left[f(\tilde{x}^k) - f(x^*)\right] \leq \bar{\theta}^k \left[f(x^0) - f(x^*)\right].$$

We emphasize that there exist values of the parameters for which the conditions in Theorem 1 and Corollary 1 are easily satisfied. For instance, setting $\eta = 1/16(\lambda n + L)$, $\kappa = 4/\lambda\eta$, $\beta = (2\lambda n + L)/L$ and $c = 2/\eta n$, the conditions in Theorem 1 are satisfied for sufficiently large $m$. Additionally, in the high condition number regime of $L/\lambda = n$, we can obtain constant $\theta < 1$ (say 0.5) with $m = O(n)$ epoch size (similar to [6, 10]). This leads to a computational complexity of $O(n \log(1/\epsilon))$ for HSAG to achieve $\epsilon$ accuracy in the objective function as opposed to $O(n^2 \log(1/\epsilon))$ for batch gradient descent method. Please refer to the appendix for more details on the parameters in Theorem 1.

## 3 Asynchronous Stochastic Variance Reduction

We are now ready to present asynchronous versions of the algorithms captured by our general framework. We first describe our setup before delving into the details of these algorithms. Our model of computation is similar to the ones used in Hogwild! [21] and AsySCD [14]. We assume a multicore architecture where each core makes stochastic gradient updates to a centrally stored vector $x$ in an asynchronous manner. There are four key components in our asynchronous algorithm; these are briefly described below.

1. **Read**: Read the iterate $x$ and compute the gradient $\nabla f_{i_t}(x)$ for a randomly chosen $i_t$.
2. **Read schedule iterate**: Read the schedule iterate $A$ and compute the gradients required for update in Algorithm 1.
3. **Update**: Update the iterate $x$ with the computed incremental update in Algorithm 1.
4. **Schedule Update**: Run a scheduler update for updating $A$.

Each processor repeatedly runs these procedures concurrently, without any synchronization. Hence, $x$ may change in between Step 1 and Step 3. Similarly, $A$ may change in between Steps 2 and 4. In fact, the states of iterates $x$ and $A$ can correspond to different time-stamps. We maintain a global counter $t$ to track the number of updates successfully executed. We use $D(t) \in [t]$ and $D'(t) \in [t]$ to denote the particular $x$-iterate and $A$-iterate used for evaluating the update at the $t^{\text{th}}$ iteration. We assume that the delay in between the time of evaluation and updating is bounded by a non-negative integer $\tau$, i.e., $t - D(t) \leq \tau$ and $t - D'(t) \leq \tau$. The bound on the staleness captures the degree of parallelism in the method: such parameters are typical in asynchronous systems (see e.g., [3, 14]). Furthermore, we also assume that the system is synchronized after every epoch i.e., $D(t) \geq km$ for $t \geq km$. We would like to emphasize that the assumption is not strong since such a synchronization needs to be done only once per epoch.

For the purpose of our analysis, we assume a consistent read model. In particular, our analysis assumes that the vector $x$ used for evaluation of gradients is a valid iterate that existed at some point in time. Such an assumption typically amounts to using locks in practice. This problem can be avoided by using random coordinate updates as in [21] (see Section 4 of [21]) but such a procedure is computationally wasteful in practice. We leave the analysis of inconsistent read model as future work. Nonetheless, we report results for both locked and lock-free implementations (see Section 4).

### 3.1 Convergence Analysis

The key ingredients to the success of asynchronous algorithms for multicore stochastic gradient descent are sparsity and "disjointness" of the data matrix [21]. More formally, suppose $f_i$ only depends on $x_{e_i}$ where $e_i \subseteq [d]$ i.e., $f_i$ acts only on the components of $x$ indexed by the set $e_i$. Let $\|x\|_i^2$ denote $\sum_{j \in e_i} \|x_j\|^2$; then, the convergence depends on $\Delta$, the smallest constant such that $\mathbb{E}_i[\|x\|_i^2] \leq \Delta \|x\|^2$. Intuitively, $\Delta$ denotes the average frequency with which a feature appears in the data matrix. We are interested in situations where $\Delta \ll 1$. As a warm up, let us first discuss convergence analysis for asynchronous SVRG. The general case is similar, but much more involved. Hence, it is instructive to first go through the analysis of asynchronous SVRG.

**Theorem 2.** *Suppose step size $\eta$, epoch size $m$ are chosen such that the following condition holds:*

$$0 < \theta_s := \frac{\left(\frac{1}{\lambda \eta m} + 4L\left(\frac{\eta + L\Delta\tau^2\eta^2}{1 - 2L^2\Delta\eta^2\tau^2}\right)\right)}{\left(1 - 4L\left(\frac{\eta + L\Delta\tau^2\eta^2}{1 - 2L^2\Delta\eta^2\tau^2}\right)\right)} < 1.$$

*Then, for the iterates of an asynchronous variant of Algorithm 1 with SVRG schedule and probabilities $p_i = 1/m$ for all $i \in [m]$, we have*

$$\mathbb{E}[f(\tilde{x}^{k+1}) - f(x^*)] \leq \theta_s \, \mathbb{E}[f(\tilde{x}^k) - f(x^*)].$$

The bound obtained in Theorem 2 is useful when $\Delta$ is small. To see this, as earlier, consider the indicative case where $L/\lambda = n$. The synchronous version of SVRG obtains a convergence rate of $\theta = 0.5$ for step size $\eta = 0.1/L$ and epoch size $m = O(n)$. For the asynchronous variant of SVRG, by setting $\eta = 0.1/2(\max\{1, \Delta^{1/2}\tau\}L)$, we obtain a similar rate with $m = O(n + \Delta^{1/2}\tau n)$. To obtain this, set $\eta = \rho/L$ where $\rho = 0.1/2(\max\{1, \Delta^{1/2}\tau\})$ and $\theta_s = 0.5$. Then, a simple calculation gives the following:

$$\frac{m}{n} = \frac{2}{\rho}\left(\frac{1 - 2\Delta\tau^2\rho^2}{1 - 12\rho - 14\Delta\tau^2\rho^2}\right) \leq c'\max\{1, \Delta^{1/2}\tau\},$$

where $c'$ is some constant. This follows from the fact that $\rho = 0.1/2(\max\{1, \Delta^{1/2}\tau\})$. Suppose $\tau < 1/\Delta^{1/2}$. Then we can achieve nearly the same guarantees as the synchronous version, but $\tau$ times faster since we are running the algorithm asynchronously. For example, consider the sparse setting where $\Delta = o(1/n)$; then it is possible to get near linear speedup when $\tau = o(n^{1/2})$. On the other hand, when $\Delta^{1/2}\tau > 1$, we can obtain a theoretical speedup of $1/\Delta^{1/2}$.

We finally provide the convergence result for the asynchronous algorithm in the general case. The proof is complicated by the fact that set $A$, unlike in SVRG, changes during the epoch. The key idea is that only a single element of $A$ changes at each iteration. Furthermore, it can only change to one of the iterates in the epoch. This control provides a handle on the error obtained due to the staleness. Due to space constraints, the proof is relegated to the appendix.

**Theorem 3.** *For any positive parameters $c, \beta, \kappa > 1$, step size $\eta$ and epoch size $m$, we define the following quantities:*

$$\zeta = \left(c\eta^2 + \left(1 - \frac{1}{\kappa}\right)^{-\tau} cL\Delta\tau^2\eta^3\right),$$

$$\gamma_a = \kappa\left[1 - \left(1 - \frac{1}{\kappa}\right)^m\right]\left[2c\eta - 8\zeta L(1+\beta) - \frac{2c}{\kappa\lambda} - \frac{96\zeta L\tau}{n}\left(1 - \frac{1}{\kappa}\right)^{-\tau} - \frac{1}{n}\right],$$

$$\theta_a = \max\left\{\left[\frac{2c}{\gamma_a\lambda}\left(1 - \frac{1}{\kappa}\right)^m + \frac{8\zeta L\left(1 + \frac{1}{\beta}\right)}{\gamma_a}\kappa\left[1 - \left(1 - \frac{1}{\kappa}\right)^m\right]\right], \left(1 - \frac{1}{\kappa}\right)^m\right\}.$$

*Suppose probabilities $p_i \propto (1 - \frac{1}{\kappa})^{m-i}$, parameters $\beta, \kappa$, step-size $\eta$, and epoch size $m$ are chosen such that the following conditions are satisfied:*

$$\frac{1}{\kappa} + 8\zeta L\left(1 + \frac{1}{\beta}\right) + \frac{96\zeta L\tau}{n}\left(1 - \frac{1}{\kappa}\right)^{-\tau} \leq \frac{1}{n}, \quad \eta^2 \leq \left(1 - \frac{1}{\kappa}\right)^{m-1}\frac{1}{12L^2\Delta\tau^2}, \quad \gamma_a > 0, \quad \theta_a < 1.$$

*Then, for the iterates of asynchronous variant of Algorithm 1 with HSAG schedule we have*

$$\mathbb{E}\left[f(\tilde{x}^{k+1}) - f(x^*) + \frac{1}{\gamma_a}\tilde{G}_{k+1}\right] \leq \theta_a\mathbb{E}\left[f(\tilde{x}^k) - f(x^*) + \frac{1}{\gamma_a}\tilde{G}_k\right].$$

**Corollary 2.** *Note that $\tilde{G}_k \geq 0$ and therefore, under the conditions specified in Theorem 3 and with $\bar{\theta}_a = \theta_a(1 + 1/\gamma_a) < 1$, we have*

$$\mathbb{E}\left[f(\tilde{x}^k) - f(x^*)\right] \leq \bar{\theta}_a^k\left[f(x^0) - f(x^*)\right].$$

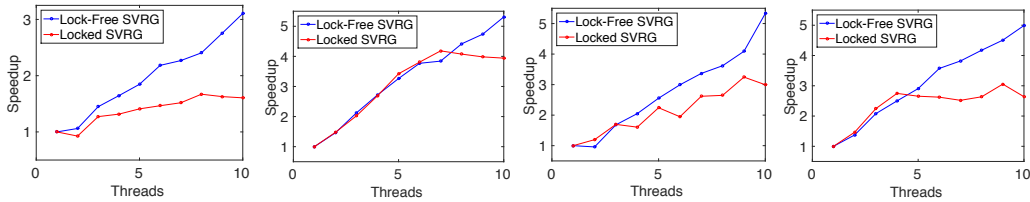

Figure 3: $l_2$-regularized logistic regression. Speedup curves for Lock-Free SVRG and Locked SVRG on rcv1 (left), real-sim (left center), news20 (right center) and url (right) datasets. We report the speedup achieved by increasing the number of threads.

By using step size normalized by $\Delta^{1/2}\tau$ (similar to Theorem 2) and parameters similar to the ones specified after Theorem 1 we can show speedups similar to the ones obtained in Theorem 2. Please refer to the appendix for more details on the parameters in Theorem 3.

Before ending our discussion on the theoretical analysis, we would like to highlight an important point. Our emphasis throughout the paper was on generality. While the results are presented here in full generality, one can obtain stronger results in specific cases. For example, in the case of SAGA, one can obtain *per iteration* convergence guarantees (see [6]) rather than those corresponding to *per epoch* presented in the paper. Also, SAGA can be analyzed without any additional synchronization per epoch. However, there is no qualitative difference in these guarantees accumulated over the epoch. Furthermore, in this case, our analysis for both synchronous and asynchronous cases can be easily modified to obtain convergence properties similar to those in [6].

## 4   Experiments

We present our empirical results in this section. For our experiments, we study the problem of binary classification via $l_2$-regularized logistic regression. More formally, we are interested in the following optimization problem:

$$\min_x \frac{1}{n} \sum_{i=1}^{n} \left( \log(1 + \exp(-y_i z_i^\top x)) + \lambda \|x\|^2 \right), \tag{4.1}$$

where $z_i \in \mathbb{R}^d$ and $y_i$ is the corresponding label for each $i \in [n]$. In all our experiments, we set $\lambda = 1/n$. Note that such a choice leads to high condition number.

A careful implementation of SVRG is required for sparse gradients since the implementation as stated in Algorithm 1 will lead to dense updates at each iteration. For an efficient implementation, a scheme like the 'just-in-time' update scheme, as suggested in [25], is required. Due to lack of space, we provide the implementation details in the appendix.

We evaluate the following algorithms for our experiments:

- **Lock-Free SVRG**: This is the lock-free asynchronous variant of Algorithm 1 using SVRG schedule; all threads can read and update the parameters with any synchronization. Parameter updates are performed through atomic *compare-and-swap* instruction [21]. A constant step size that gives the best convergence is chosen for the dataset.

- **Locked SVRG**: This is the locked version of the asynchronous variant of Algorithm 1 using SVRG schedule. In particular, we use a *concurrent read exclusive write* locking model, where all threads can read the parameters but only one threads can update the parameters at a given time. The step size is chosen similar to Lock-Free SVRG.

- **Lock-Free SGD**: This is the lock-free asynchronous variant of the SGD algorithm (see [21]). We compare two different versions of this algorithm: (i) SGD with constant step size (referred to as CSGD). (ii) SGD with decaying step size $\eta_0 \sqrt{\sigma_0/(t + \sigma_0)}$ (referred to as DSGD), where constants $\eta_0$ and $\sigma_0$ specify the scale and speed of decay. For each of these versions, step size is tuned for each dataset to give the best convergence progress.

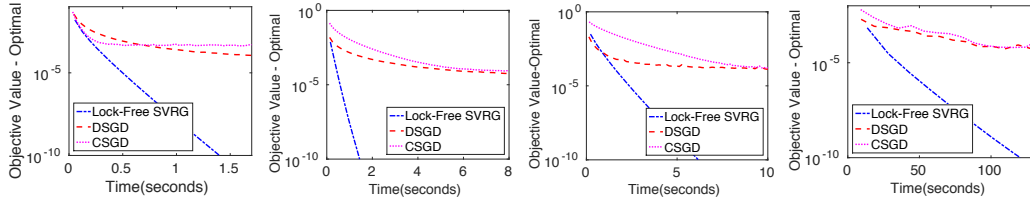

Figure 4: $l_2$-regularized logistic regression. Training loss residual $f(x) - f(x^*)$ versus time plot of Lock-Free SVRG, DSGD and CSGD on rcv1 (left), real-sim (left center), news20 (right center) and url (right) datasets. The experiments are parallelized over 10 cores.

All the algorithms were implemented in C++ [2]. We run our experiments on datasets from LIBSVM website[3]. Similar to [29], we normalize each example in the dataset so that $\|z_i\|_2 = 1$ for all $i \in [n]$. Such a normalization leads to an upper bound of $0.25$ on the Lipschitz constant of the gradient of $f_i$. The epoch size $m$ is chosen as $2n$ (as recommended in [10]) in all our experiments. In the first experiment, we compare the speedup achieved by our asynchronous algorithm. To this end, for each dataset we first measure the time required for the algorithm to each an accuracy of $10^{-10}$ (i.e., $f(x) - f(x^*) < 10^{-10}$). The speedup with $P$ threads is defined as the ratio of the runtime with a single thread to the runtime with $P$ threads. Results in Figure 3 show the speedup on various datasets. As seen in the figure, we achieve significant speedups for all the datasets. Not surprisingly, the speedup achieved by Lock-free SVRG is much higher than ones obtained by locking. Furthermore, the lowest speedup is achieved for rcv1 dataset. Similar speedup behavior was reported for this dataset in [21]. It should be noted that this dataset is not sparse and hence, is a bad case for the algorithm (similar to [21]).

For the second set of experiments we compare the performance of Lock-Free SVRG with stochastic gradient descent. In particular, we compare with the variants of stochastic gradient descent, DSGD and CSGD, described earlier in this section. It is well established that the performance of variance reduced stochastic methods is better than that of SGD. We would like to empirically verify that such benefits carry over to the asynchronous variants of these algorithms. Figure 4 shows the performance of Lock-Free SVRG, DSGD and CSGD. Since the computation complexity of each epoch of these algorithms is different, we directly plot the objective value versus the runtime for each of these algorithms. We use 10 cores for comparing the algorithms in this experiment. As seen in the figure, Lock-Free SVRG outperforms both DSGD and CSGD. The performance gains are qualitatively similar to those reported in [10] for the synchronous versions of these algorithms. It can also be seen that the DSGD, not surprisingly, outperforms CSGD in all the cases. In our experiments, we observed that Lock-Free SVRG, in comparison to SGD, is relatively much less sensitive to the step size and more robust to increasing threads.

# 5   Discussion & Future Work

In this paper, we presented a unifying framework based on [6], that captures many popular variance reduction techniques for stochastic gradient descent. We use this framework to develop a simple hybrid variance reduction method. The primary purpose of the framework, however, was to provide a common platform to analyze various variance reduction techniques. To this end, we provided convergence analysis for the framework under certain conditions. More importantly, we propose an asynchronous algorithm for the framework with provable convergence guarantees. The key consequence of our approach is that we obtain asynchronous variants of several algorithms like SVRG, SAGA and S2GD. Our asynchronous algorithms exploits sparsity in the data to obtain near linear speedup in settings that are typically encountered in machine learning.

For future work, it would be interesting to perform an empirical comparison of various schedules. In particular, it would be worth exploring the space-time-accuracy tradeoffs of these schedules. We would also like to analyze the effect of these tradeoffs on the asynchronous variants.

**Acknowledgments.** SS was partially supported by NSF IIS-1409802.

## Footnotes

[1]Though we should note that SGD also applies to the harder stochastic optimization problem $\min F(x) = \mathbb{E}[f(x; \xi)]$, which need not be a finite-sum.

[2]All experiments were conducted on a Google Compute Engine n1-highcpu-32 machine with 32 processors and 28.8 GB RAM.

[3]http://www.csie.ntu.edu.tw/~cjlin/libsvmtools/datasets/binary.html

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
