[Supplementary Material]

# A  Appendix

**Notation**: We use $D_f$ to denote the Bregman divergence (defined below) for function $f$.

$$D_f(x, y) = f(x) - f(y) - \langle \nabla f(y), x - y \rangle.$$

For ease of exposition, we use $\mathbb{E}[X]$ to denote the expectation the random variable $X$ with respect to indices $\{i_1, \ldots, i_t\}$ when $X$ depends on just these indices up to step $t$. This dependence will be clear from the context. We use $\mathbb{1}$ to denote the indicator function. We assume $\sum_{d=i}^{j} a_d = 0$ if $i > j$.

We would like to clarify the definition of $x^{km}$ here. As noted in the main text, we assume that $x^{km+m}$ is replaced with an element chosen randomly from $\{x^{km}, \ldots, x^{km+m-1}\}$ with probability $\{p_1, \cdots, p_m\}$ at the end of the $(k+1)^{\text{th}}$ epoch. However, whenever $x^{km}$ appears in the analysis (proofs), it represents the iterate before this replacement.

## Implementation Details

Since we are interested in sparse datasets, simply taking $f_i(x) = \log(1 + \exp(-y_i z_i^\top x)) + \lambda \|x\|^2$ is not efficient as it requires updating the whole vector $x$ at each iteration. This is due to the regularization term in each of the $f_i$'s. Instead, similar to [21], we rewrite problem in (4.1) as follows:

$$\min_x \frac{1}{n} \sum_{i=1}^{n} \left( \log(1 + \exp(-y_i z_i^\top x)) + \lambda \sum_{j \in nz(z_i)} \frac{\|x_j\|^2}{d_j} \right),$$

where $nz(z)$ represents the non-zero components of vector $z$, and $d_j = \sum_i \mathbb{1}(j \in nz(z_i))$. While this leads to sparse gradients at each iteration, updates in SVRG are still dense due to the part of the update that contains $\sum_i \nabla f_i(\alpha_i)/n$. This problem can be circumvented by using the following update scheme. First, recall that for SVRG, $\sum_i \nabla f_i(\alpha_i)/n$ does not change during an epoch (see Figure 1). Therefore, during the $(k+1)^{\text{st}}$ epoch we have the following relationship:

$$x^t = \left[ \tilde{x}^k - \eta \sum_{j=km}^{t-1} (\nabla f_{i_j}(x^j) - \nabla f_{i_j}(\tilde{x}^k)) \right] - \left[ \frac{\eta \nu_t}{n} \sum_{i=1}^{n} \nabla f_i(\tilde{x}^k) \right],$$

where $\nu_t = (t - km)$. Each step of the algorithm involves updating the bracketed terms separately. The updates to the first term in the above equation are sparse while those to the second term are just simple scalar additions, since we already maintain the average gradient $\sum_{i=1}^{n} \nabla f_i(\tilde{x}^k)/n$. When the gradient of $f_{i_t}$ at $x^t$ is needed, we *only* calculate components of $x^t$ required for $f_{i_t}$ on the fly by aggregating these two terms. Hence, each step of this update procedure can be implemented in a way that respects sparsity of the data.

## Proof of Theorem 1

*Proof.* We expand function $f$ as $f(x) = g(x) + h(x)$ where $g(x) = \frac{1}{n} \sum_{i \in S} f_i(x)$ and $h(x) = \frac{1}{n} \sum_{i \notin S} f_i(x)$. Let the present epoch be $k + 1$. We define the following:

$$v^t = \frac{1}{\eta}(x^{t+1} - x^t) = -\left[ \nabla f_{i_t}(x^t) - \nabla f_{i_t}(\alpha_{i_t}^t) + \frac{1}{n} \sum_i \nabla f_i(\alpha_i^t) \right]$$

$$G_t = \frac{1}{n} \sum_{i \in S} \left( f_i(\alpha_i^t) - f_i(x^*) - \langle \nabla f_i(x^*), \alpha_i^t - x^* \rangle \right)$$

$$R_t = \mathbb{E} \left[ c\|x^t - x^*\|^2 + G_t \right].$$

We first observe that $\mathbb{E}[v^t] = -\nabla f(x^t)$. This follows from the unbiasedness of the gradient at each iteration. Using this observation, we have the following:

$$\mathbb{E}[R_{t+1}] = \mathbb{E}[c\|x^{t+1} - x^*\|^2 + G_{t+1}] = \mathbb{E}[c\|x^t + \eta v^t - x^*\|^2 + G_{t+1}]$$

$$= c\mathbb{E} \left[ \|x^t - x^*\|^2 \right] + c\eta^2 \mathbb{E} \left[ \|v^t\|^2 \right] + 2c\eta \mathbb{E} \left[ \langle x^t - x^*, v^t \rangle \right] + \mathbb{E}[G_{t+1}]$$

$$\leq c\mathbb{E} \left[ \|x^t - x^*\|^2 \right] + c\eta^2 \mathbb{E} \left[ \|v^t\|^2 \right] - 2c\eta \mathbb{E} \left[ f(x^t) - f(x^*) \right] + \mathbb{E}[G_{t+1}]. \tag{A.1}$$

The last step follows from convexity of $f$ and the unbiasedness of $v^t$. We have the following relationship between $G_{t+1}$ and $G_t$.

$$
\begin{aligned}
\mathbb{E}[G_{t+1}] &= \left(1 - \frac{1}{n}\right)\mathbb{E}\left[G_t\right] + \frac{1}{n}\mathbb{E}\left[\frac{1}{n}\sum_{i\in S}\left(f_i(x^t) - f_i(x^*) - \langle\nabla f_i(x^*), x^t - x^*\rangle\right)\right] \\
&= \left(1 - \frac{1}{n}\right)\mathbb{E}\left[G_t\right] + \frac{1}{n}\mathbb{E}[D_g(x^t, x^*)].
\end{aligned}
\tag{A.2}
$$

This follows from the definition of the schedule of HSAG for indices in $S$. Substituting the above relationship in Equation (A.1) we get the following.

$$
\begin{aligned}
R_{t+1} &\leq R_t + c\eta^2\mathbb{E}\left[\|v^t\|^2\right] - 2c\eta\mathbb{E}\left[f(x^t) - f(x^*)\right] - \frac{1}{n}\mathbb{E}[G_t] + \frac{1}{n}\mathbb{E}[D_g(x^t, x^*)] \\
&\leq \left(1 - \frac{1}{\kappa}\right)R_t + \frac{c}{\kappa}\mathbb{E}[\|x^t - x^*\|^2] + c\eta^2\mathbb{E}\left[\|v^t\|^2\right] - 2c\eta\mathbb{E}\left[f(x^t) - f(x^*)\right] \\
&\quad + \left(\frac{1}{\kappa} - \frac{1}{n}\right)\mathbb{E}[G_t] + \frac{1}{n}\mathbb{E}[D_g(x^t, x^*)] \\
&:= \left(1 - \frac{1}{\kappa}\right)R_t + b_t.
\end{aligned}
$$

We describe the bounds for $b_t$ (defined below).

$$
\begin{aligned}
b_t = \frac{c}{\kappa}\underbrace{\mathbb{E}[\|x^t - x^*\|^2]}_{T_1} + c\eta^2\underbrace{\mathbb{E}\left[\|v^t\|^2\right]}_{T_2} - 2c\eta\mathbb{E}\left[f(x^t) - f(x^*)\right] \\
+ \left(\frac{1}{\kappa} - \frac{1}{n}\right)\mathbb{E}[G_t] + \frac{1}{n}\mathbb{E}[D_g(x^t, x^*)].
\end{aligned}
$$

The terms $T_1$ and $T2$ can be bounded in the following fashion:

$$
\begin{aligned}
T_1 &= \mathbb{E}[\|x^t - x^*\|^2] \leq \frac{2}{\lambda}\mathbb{E}[f(x^t) - f(x^*)] \\
T_2 &= \mathbb{E}\left[\|v^t\|^2\right] \leq \left(1 + \frac{1}{\beta}\right)\mathbb{E}\left[\|\nabla f_{i_t}(\alpha_{i_t}^t) - \nabla f_{i_t}(x^*)\|^2\right] + (1 + \beta)\mathbb{E}\left[\|\nabla f_{i_t}(x^t) - \nabla f_{i_t}(x^*)\|^2\right] \\
&\leq \frac{2L}{n}\left(1 + \frac{1}{\beta}\right)\mathbb{E}\sum_i\left[f_i(\alpha_i^t) - f(x^*) - \langle\nabla f_i(x^*), \alpha_i^t - x^*\rangle\right] \\
&\quad + \frac{2L}{n}(1 + \beta)\mathbb{E}\sum_i\left[f_i(x^t) - f(x^*)\right] \\
&\leq 2L\left(1 + \frac{1}{\beta}\right)\mathbb{E}\left[G_t + D_h(\tilde{x}^k, x^*)\right] + 2L(1 + \beta)\mathbb{E}[f(x^t) - f(x^*)].
\end{aligned}
$$

The bound on $T_1$ is due to strong convexity nature of function $f$. The first inequality and second inequalities on $T_2$ directly follows from Lemma 3 of [6] and simple application of Lemma 1 respectively. The third inequality follows from the definition of $G_t$ and the fact that $\alpha_i^t = \tilde{x}^k$ for all $i \notin S$ and $t \in \{km, \ldots, km + m - 1\}$.

Substituting these bounds $T_1$ and $T_2$ in $b_t$, we get

$$
\begin{aligned}
b_t \leq &- \left[ 2c\eta - 2cL\eta^2(1+\beta) - \frac{2c}{\kappa\lambda} \right] \mathbb{E}\left[ f(x^t) - f(x^*) \right] \\
&+ \left( \frac{1}{\kappa} + 2cL\eta^2 \left( 1 + \frac{1}{\beta} \right) - \frac{1}{n} \right) \mathbb{E}[G_t] + \frac{1}{n} \mathbb{E}[D_g(x^t, x^*)] \\
&+ 2cL\eta^2 \left( 1 + \frac{1}{\beta} \right) \mathbb{E}\left[ D_h(\tilde{x}^k, x^*) \right] \\
\leq &- \left[ 2c\eta - 2cL\eta^2(1+\beta) - \frac{1}{n} - \frac{2c}{\kappa\lambda} \right] \mathbb{E}\left[ f(x^t) - f(x^*) \right] \\
&+ \left( \frac{1}{\kappa} + 2cL\eta^2 \left( 1 + \frac{1}{\beta} \right) - \frac{1}{n} \right) \mathbb{E}[G_t] + 2cL\eta^2 \left( 1 + \frac{1}{\beta} \right) \mathbb{E}\left[ D_h(\tilde{x}^k, x^*) \right] \\
\leq &- \left[ 2c\eta - 2cL\eta^2(1+\beta) - \frac{1}{n} - \frac{2c}{\kappa\lambda} \right] \mathbb{E}\left[ f(x^t) - f(x^*) \right] + 2cL\eta^2 \left( 1 + \frac{1}{\beta} \right) \mathbb{E}\left[ D_h(\tilde{x}^k, x^*) \right].
\end{aligned}
$$

$$(A.3)$$

The second inequality follows from Lemma 2. In particular, we use the fact that $f(x) - f(x^*) = D_f(x, x^*)$ and $D_f(x, x^*) = D_g(x, x^*) + D_h(x, x^*) \geq D_g(x, x^*)$. The third inequality follows from the following for the choice of our parameters:

$$
\frac{1}{\kappa} + 2Lc\eta^2 \left( 1 + \frac{1}{\beta} \right) \leq \frac{1}{n}.
$$

Applying the recursive relationship on $R_{t+1}$ for m iterations, we get

$$
R_{km+m} \leq \left( 1 - \frac{1}{\kappa} \right)^m \tilde{R}_k + \sum_{j=0}^{m-1} \left( 1 - \frac{1}{\kappa} \right)^{m-1-j} b_{km+j}
$$

where

$$
\tilde{R}_k = \mathbb{E}\left[ c\|\tilde{x}^k - x^*\|^2 + \tilde{G}_k \right].
$$

Substituting the bound on $b_t$ from Equation (A.3) in the above equation we get the following inequality:

$$
\begin{aligned}
R_{km+m} \leq &\left( 1 - \frac{1}{\kappa} \right)^m \tilde{R}_k \\
&- \sum_{j=0}^{m-1} \left( 2c\eta(1 - L\eta(1+\beta)) - \frac{1}{n} - \frac{2c}{\kappa\lambda} \right) \left( 1 - \frac{1}{\kappa} \right)^{m-1-j} \mathbb{E}\left[ f(x^{km+j}) - f(x^*) \right] \\
&+ \sum_{j=0}^{m-1} \left( 1 - \frac{1}{\kappa} \right)^{m-1-j} 2Lc\eta^2 \left( 1 + \frac{1}{\beta} \right) \mathbb{E}\left[ h(\tilde{x}^k) - h(x^*) - \langle \nabla h(x^*), \tilde{x}^k - x^* \rangle \right].
\end{aligned}
$$

We now use the fact that $\tilde{x}^{k+1}$ is chosen randomly from $\{x^{km}, \ldots, x^{km+m-1}\}$ with probabilities proportional to $\{(1 - 1/\kappa)^{m-1}, \ldots, 1\}$ we have the following consequence of the above inequality.

$$
\begin{aligned}
&R_{km+m} + \kappa \left[ 1 - \left( 1 - \frac{1}{\kappa} \right)^m \right] \left( 2c\eta(1 - L\eta(1+\beta)) - \frac{1}{n} - \frac{2c}{\kappa\lambda} \right) \mathbb{E}\left[ f(\tilde{x}^{k+1}) - f(x^*) \right] \\
&\leq \frac{2c}{\lambda} \left( 1 - \frac{1}{\kappa} \right)^m \mathbb{E}\left[ f(\tilde{x}^k) - f(x^*) \right] + \left( 1 - \frac{1}{\kappa} \right)^m \mathbb{E}\left[ \tilde{G}_k \right] \\
&\quad + 2Lc\eta^2 \kappa \left[ 1 - \left( 1 - \frac{1}{\kappa} \right)^m \right] \left( 1 + \frac{1}{\beta} \right) \mathbb{E}\left[ D_h(\tilde{x}^k, x^*) \right].
\end{aligned}
$$

For obtaining the above inequality, we used the strongly convex nature of function $f$. Again, using the Bregman divergence based inequality (see Lemma 2)

$$
f(x) - f(x^*) = D_f(x, x^*) = D_g(x, x^*) + D_h(x, x^*) \geq D_h(x, x^*),
$$

we have the following inequality

$$R_{km+m} + \kappa \left[1 - \left(1 - \frac{1}{\kappa}\right)^m\right] \left(2c\eta(1 - L\eta(1+\beta)) - \frac{1}{n} - \frac{2c}{\kappa\lambda}\right) \mathbb{E}\left[f(\tilde{x}^{k+1}) - f(x^*)\right]$$

$$\leq \left[\frac{2c}{\lambda}\left(1 - \frac{1}{\kappa}\right)^m + 2Lc\eta^2\kappa\left(1 + \frac{1}{\beta}\right)\left[1 - \left(1 - \frac{1}{\kappa}\right)^m\right]\right] \mathbb{E}\left[f(\tilde{x}^k) - f(x^*)\right] + \left(1 - \frac{1}{\kappa}\right)^m \mathbb{E}\left[\tilde{G}_k\right].$$

(A.4)

We use the following notation:

$$\gamma = \kappa\left[1 - \left(1 - \frac{1}{\kappa}\right)^m\right]\left(2c\eta(1 - L\eta(1+\beta)) - \frac{1}{n} - \frac{2c}{\kappa\lambda}\right)$$

$$\theta = \max\left\{\left[\frac{2c}{\gamma\lambda}\left(1 - \frac{1}{\kappa}\right)^m + \frac{2Lc\eta^2}{\gamma}\left(1 + \frac{1}{\beta}\right)\kappa\left[1 - \left(1 - \frac{1}{\kappa}\right)^m\right]\right], \left(1 - \frac{1}{\kappa}\right)^m\right\}.$$

Using the above notation, we have the following inequality from Equation (A.4).

$$\mathbb{E}\left[f(\tilde{x}^{k+1}) - f(x^*) + \frac{1}{\gamma}\tilde{G}_{k+1}\right] \leq \theta\, \mathbb{E}\left[f(\tilde{x}^k) - f(x^*) + \frac{1}{\gamma}\tilde{G}_k\right],$$

where $\theta < 1$ is a constant that depends on the parameters used in the algorithm. □

## Proof of Theorem 2

*Proof.* Let the present epoch be $k + 1$. Recall that $D(t)$ denotes the iterate used in the $t^{\text{th}}$ iteration of the algorithm. We define the following:

$$u^t = -\left[\nabla f_{i_t}(x^{D(t)}) - \nabla f_{i_t}(\tilde{x}^k) + \nabla f(\tilde{x}^k)\right]$$

$$v^t = -\left[\nabla f_{i_t}(x^t) - \nabla f_{i_t}(\tilde{x}^k) + \nabla f(\tilde{x}^k)\right].$$

We have the following:

$$\mathbb{E}\|x^{t+1} - x^*\|^2 = \mathbb{E}\|x^t + \eta u^t - x^*\|^2 = \mathbb{E}\left[\|x^t - x^*\|^2 + \eta^2\|u^t\|^2 + 2\eta\langle x^t - x^*, u^t\rangle\right]. \quad \text{(A.5)}$$

We first bound the last term of the above inequality. We expand the term in the following manner:

$$\mathbb{E}\langle x^t - x^*, u^t\rangle = \mathbb{E}\left[\langle x^* - x^t, \nabla f_{i_t}(x^{D(t)})\rangle\right]$$

$$= \underbrace{\mathbb{E}\left[\langle x^* - x^{D(t)}, \nabla f_{i_t}(x^{D(t)})\rangle\right]}_{T_3}$$

$$+ \underbrace{\sum_{d=D(t)}^{t-1} \mathbb{E}\left[\langle x^d - x^{d+1}, \nabla f_{i_t}(x^d)\rangle\right]}_{T_4} + \underbrace{\sum_{d=D(t)}^{t-1} \mathbb{E}\left[\langle x^d - x^{d+1}, \nabla f_{i_t}(x^{D(t)}) - \nabla f_{i_t}(x^d)\rangle\right]}_{T_5}.$$

(A.6)

The first equality directly follows from the definition of $u^t$ and its property of unbiasedness. The second step follows from simple algebraic calculations. Terms $T_3$ and $T_4$ can be bounded in the following way:

$$T_3 \leq \mathbb{E}[f_{i_t}(x^*) - f_{i_t}(x^{D(t)})]. \quad \text{(A.7)}$$

This bound directly follows from convexity of function $f_{i_t}$.

$$T_4 = \sum_{d=D(t)}^{t-1} \mathbb{E}\left[\langle x^d - x^{d+1}, \nabla f_{i_t}(x^d)\rangle\right]$$

$$\leq \sum_{d=D(t)}^{t-1} \mathbb{E}\left[f_{i_t}(x^d) - f_{i_t}(x^{d+1}) + \frac{L}{2}\|x^{d+1} - x^d\|_{i_t}^2\right]$$

$$\leq \mathbb{E}\left[f_{i_t}(x^{D(t)}) - f_{i_t}(x^t)\right] + \frac{L\Delta}{2}\sum_{d=D(t)}^{t-1} \mathbb{E}\left[\|x^{d+1} - x^d\|^2\right]. \quad \text{(A.8)}$$

The first inequality follows from lipschitz continuous nature of the gradient of function $f_{i_t}$. The second inequality follows from the definition of $\Delta$. The last term $T_5$ can be bounded in the following manner.

$$T_5 = \mathbb{E}\left[ \sum_{d=D(t)}^{t-1} \langle x^d - x^{d+1}, \nabla f_{i_t}(x^{D(t)}) - \nabla f_{i_t}(x^d) \rangle \right]$$

$$\leq \mathbb{E}\left[ \sum_{d=D(t)}^{t-1} \|x^{d+1} - x^d\|_{i_t} \|\nabla f_{i_t}(x^{D(t)}) - \nabla f_{i_t}(x^d)\| \right]$$

$$\leq \mathbb{E}\left[ \sum_{d=D(t)}^{t-1} \|x^{d+1} - x^d\|_{i_t} \sum_{j=D(t)}^{d-1} \|\nabla f_{i_t}(x^{j+1}) - \nabla f_{i_t}(x^j)\| \right]$$

$$\leq \mathbb{E}\left[ \sum_{d=D(t)}^{t-1} \sum_{j=D(t)}^{d-1} \frac{L}{2} \left( \|x^{d+1} - x^d\|_{i_t}^2 + \|x^{j+1} - x^j\|_{i_t}^2 \right) \right]$$

$$\leq \frac{L\Delta(\tau-1)}{2} \mathbb{E} \sum_{d=D(t)}^{t-1} \|x^{d+1} - x^d\|^2. \tag{A.9}$$

The first inequality follows from Cauchy-Schwartz inequality. The second inequality follows from repeated application of triangle inequality. The third step is a simple application of AM-GM inequality and the fact that gradient of the function $f_{i_t}$ is lipschitz continuous. Finally, the last step can be obtained by using a simple counting argument, the fact that the staleness in gradient is at most $\tau$ and the definition of $\Delta$.

By combining the bounds on $T_3, T_4$ and $T_5$ in Equations (A.7), (A.8) and (A.9) respectively and substituting the sum in Equation (A.6), we get

$$\mathbb{E}\langle x^t - x^*, u^t \rangle \leq \mathbb{E}\left[ f(x^*) - f(x^t) + \frac{L\Delta\tau}{2} \sum_{d=D(t)}^{t-1} \|x^{d+1} - x^d\|^2 \right]. \tag{A.10}$$

By substituting the above inequality in Equation (A.5), we get

$$\mathbb{E}\left[ \|x^{t+1} - x^*\|^2 \right] \leq \mathbb{E}\left[ \|x^t - x^*\|^2 + \eta^2\|u^t\|^2 - 2\eta(f(x^t) - f(x^*)) + L\Delta\tau\eta^3 \sum_{d=D(t)}^{t-1} \|u^d\|^2 \right]. \tag{A.11}$$

We next bound the term $\mathbb{E}[\|u^t\|^2]$ in terms of $\mathbb{E}\left[\|v^t\|^2\right]$ in the following way:

$$\mathbb{E}\left[\|u^t\|^2\right] \leq 2\mathbb{E}\left[\|u^t - v^t\|^2 + \|v^t\|^2\right]$$

$$\leq 2\mathbb{E}\left[\|\nabla f_{i_t}(x^t) - \nabla f_{i_t}(x^{D(t)})\|^2\right] + 2\mathbb{E}\left[\|v^t\|^2\right]$$

$$\leq 2L^2\tau \sum_{d=D(t)}^{t-1} \mathbb{E}\left[\|x^{d+1} - x^d\|_{i_t}^2\right] + 2\mathbb{E}\left[\|v^t\|^2\right]$$

$$\leq 2L^2\Delta\eta^2\tau \sum_{d=D(t)}^{t-1} \mathbb{E}\left[\|u^d\|^2\right] + 2\mathbb{E}\left[\|v^t\|^2\right].$$

The first step follows from Lemma 3 for $r = 2$. The third inequality follows from the lipschitz continuous nature of the gradient and simple application of Lemma 3. Adding the above inequalities from $t = km$ to $t = km + m - 1$, we get

$$\sum_{t=km}^{km+m-1} \mathbb{E}\left[\|u^t\|^2\right] \leq \sum_{t=km}^{km+m-1} \left[ 2L^2\Delta\eta^2\tau \sum_{d=D(t)}^{t-1} \mathbb{E}\left[\|u^d\|^2\right] + 2\mathbb{E}\left[\|v^t\|^2\right] \right]$$

$$\leq 2L^2\Delta\eta^2\tau^2 \sum_{t=km}^{km+m-1} \mathbb{E}\left[\|u^t\|^2\right] + 2 \sum_{t=km}^{km+m-1} \mathbb{E}\left[\|v^t\|^2\right].$$

Here we again used a simple counting argument and the fact that the delay in the gradients is at most $\tau$. From the above inequality, we get

$$\sum_{t=km}^{km+m-1} \mathbb{E}\left[\|u^t\|^2\right] \leq \frac{2}{(1-2L^2\Delta\eta^2\tau^2)} \sum_{t=km}^{km+m-1} \mathbb{E}\left[\|v^t\|^2\right]. \tag{A.12}$$

Adding Equation (A.11) from $t = km$ to $t = km + m - 1$ and substituting Equation (A.12) in the resultant, we get

$$\mathbb{E}\left[\|x^{km+m} - x^*\|^2\right] \leq \mathbb{E}\left[\|\tilde{x}^k - x^*\|^2 + (\eta^2 + L\Delta\tau^2\eta^3)\sum_{t=km}^{km+m-1}\|u^t\|^2 - \sum_{t=km}^{km+m-1} 2\eta(f(x^t) - f(x^*))\right]$$

$$\leq \mathbb{E}\left[\|\tilde{x}^k - x^*\|^2 + 2\left(\frac{\eta^2 + L\Delta\tau^2\eta^3}{1-2L^2\Delta\eta^2\tau^2}\right)\sum_{t=km}^{km+m-1}\|v^t\|^2 - \sum_{t=km}^{km+m-1} 2\eta(f(x^t) - f(x^*))\right].$$

Here, we used the fact that the system is synchronized after every epoch. The first step follows from telescopy sum and the definition of $\tilde{x}^k$. From Lemma 3 of [6] (also see [10]), we have

$$\mathbb{E}[\|v^t\|^2] \leq 4L\mathbb{E}\left[f(x^t) - f(x^*) + f(\tilde{x}^k) - f(x^*)\right].$$

Substituting this in the inequality above, we get the following bound:

$$\left(2\eta - 8L\left(\frac{\eta^2 + L\Delta\tau^2\eta^3}{1-2L^2\Delta\eta^2\tau^2}\right)\right)m\mathbb{E}[f(\tilde{x}^{k+1}) - f(x^*)]$$

$$\leq \left(\frac{2}{\lambda} + 8L\left(\frac{\eta^2 + L\Delta\tau^2\eta^3}{1-2L^2\Delta\eta^2\tau^2}\right)m\right)\mathbb{E}[f(\tilde{x}^k) - f(x^*)].$$

$\square$

## Proof of Theorem 3

*Proof.* Let the present epoch be $k+1$. For simplicity, we assume that the iterates $x$ and $A$ used in the each iteration are from the same time step (index) i.e., $D(t) = D'(t)$ for all $t \in T$. Recall that $D(t)$ and $D'(t)$ denote the index used in the $t^{\text{th}}$ iteration of the algorithm. Our analysis can be extended to the case of $D(t) \neq D'(t)$ in a straightforward manner. We expand function $f$ as $f(x) = g(x) + h(x)$ where $g(x) = \frac{1}{n}\sum_{i\in S} f_i(x)$ and $h(x) = \frac{1}{n}\sum_{i\notin S} f_i(x)$. We define the following:

$$u^t = \frac{1}{\eta}(x^{t+1} - x^t) = -\left[\nabla f_{i_t}(x^{D(t)}) - \nabla f_{i_t}(\alpha_{i_t}^{D(t)}) + \frac{1}{n}\sum_i \nabla f_i(\alpha_i^{D(t)})\right]$$

$$v^t = -\left[\nabla f_{i_t}(x^t) - \nabla f_{i_t}(\alpha_{i_t}^t) + \frac{1}{n}\sum_i \nabla f_i(\alpha_i^t)\right].$$

We use the same Lyapunov function used in Theorem 1. We recall the following definitions:

$$G_t = \frac{1}{n}\sum_{i\in S}\left(f_i(\alpha_i^t) - f_i(x^*) - \langle\nabla f_i(x^*), \alpha_i^t - x^*\rangle\right)$$

$$R_t = \mathbb{E}\left[c\|x^t - x^*\|^2 + G_t\right].$$

Using unbiasedness of the gradient we have $\mathbb{E}[u^t] = -\nabla f(x^{D(t)})$ and $\mathbb{E}[v^t] = -\nabla f(x^t)$. Using this observation, we have the following:

$$c\mathbb{E}[\|x^{t+1} - x^*\|^2] = c\mathbb{E}[\|x^t + \eta u^t - x^*\|^2]$$

$$= c\mathbb{E}\left[\|x^t - x^*\|^2\right] + c\eta^2 \underbrace{\mathbb{E}\left[\|u^t\|^2\right]}_{T_6} + 2c\eta \underbrace{\mathbb{E}\left[\langle x^t - x^*, u^t\rangle\right]}_{T_7}. \tag{A.13}$$

We bound term $T_6$ in the following manner:

$$T_6 = \mathbb{E}\left[\|u^t\|^2\right] \leq 2\mathbb{E}\left[\|u^t - v^t\|^2\right] + 2\mathbb{E}[\|v^t\|^2]. \tag{A.14}$$

The first term can be bounded in the following manner:

$$\mathbb{E}\left[\|u^t - v^t\|^2\right] \leq \mathbb{E}\left[\left\|(\nabla f_{i_t}(x^t) - \nabla f_{i_t}(x^{D(t)})) - (\nabla f_{i_t}(\alpha_{i_t}^{D(t)}) - \nabla f_{i_t}(\alpha_{i_t}^t))\right.\right.$$

$$\left.\left. + \frac{1}{n}\sum_i (\nabla f_i(\alpha_i^t) - \nabla f_i(\alpha_i^{D(t)}))\right\|^2\right]$$

$$\leq 3\mathbb{E}\left[\left\|\nabla f_{i_t}(x^t) - \nabla f_{i_t}(x^{D(t)})\right\|^2\right] + 3\mathbb{E}\left[\left\|\nabla f_{i_t}(\alpha_{i_t}^{D(t)}) - \nabla f_{i_t}(\alpha_{i_t}^t)\right\|^2\right]$$

$$+ 3\mathbb{E}\left[\left\|\frac{1}{n}\sum_i (\nabla f_i(\alpha_i^t) - \nabla f_i(\alpha_i^{D(t)}))\right\|^2\right]$$

$$\leq 3\mathbb{E}\left[\left\|\nabla f_{i_t}(x^t) - \nabla f_{i_t}(x^{D(t)})\right\|^2\right] + 3\mathbb{E}\left[\left\|\nabla f_{i_t}(\alpha_{i_t}^{D(t)}) - \nabla f_{i_t}(\alpha_{i_t}^t)\right\|^2\right]$$

$$+ \frac{3}{n}\sum_i \mathbb{E}\left[\left\|\nabla f_i(\alpha_i^t) - \nabla f_i(\alpha_i^{D(t)})\right\|^2\right]. \tag{A.15}$$

The second step follows from Lemma 3 for r = 3. The last step follows from simple application of Jensen's inequality. The first term can be bounded easily in the following manner:

$$\mathbb{E}\left[\|\nabla f_{i_t}(x^t) - \nabla f_{i_t}(x^{D(t)})\|^2\right] \leq L^2\tau \sum_{d=D(t)}^{t-1} \mathbb{E}\left[\|x^{d+1} - x^d\|_{i_t}^2\right]$$

$$\leq L^2\Delta\eta^2\tau \sum_{d=D(t)}^{t-1} \mathbb{E}\left[\|u^d\|^2\right].$$

The second and third terms need more delicate analysis. The key insight for our analysis is that at most $\tau$ $\alpha_i$'s differ from time step $D(t)$ to $t$. This is due to the fact that the delay is bounded by $\tau$ and at most one $\alpha_i$ changes at each iteration. Furthermore, whenever there is a change in $\alpha_i$, it changes to one of the iterates $x^j$ for some $j = \{\max\{t-\tau, km\}, \ldots, t\}$. With this intuition we bound the second term in the following fashion.

$$\mathbb{E}\left[\left\|\nabla f_{i_t}(\alpha_{i_t}^{D(t)}) - \nabla f_{i_t}(\alpha_{i_t}^t)\right\|^2\right] \leq \frac{1}{n}\sum_{j=D(t)}^{t-1}\sum_{i\in S}\mathbb{E}\left[\mathbb{1}(i=i_j)\left\|\nabla f_i(x^j) - \nabla f_i(\alpha_i^{D(t)})\right\|^2\right]$$

$$\leq \frac{2}{n}\sum_{j=D(t)}^{t-1}\sum_{i\in S}\mathbb{E}\left[\mathbb{1}(i=i_j)\left(\left\|\nabla f_i(x^j) - \nabla f_i(x^*)\right\|^2 + \left\|\nabla f_i(\alpha_i^{D(t)}) - \nabla f_i(x^*)\right\|^2\right)\right]$$

$$\leq \frac{2}{n^2}\sum_{j=D(t)}^{t-1}\sum_{i\in S}\mathbb{E}\left[\left\|\nabla f_i(x^j) - \nabla f_i(x^*)\right\|^2\right] + \frac{2}{n^2}\sum_{j=D(t)}^{t-1}\sum_{i\in S}\mathbb{E}\left[\left\|\nabla f_i(\alpha_i^{D(t)}) - \nabla f_i(x^*)\right\|^2\right]$$

$$\leq \frac{4L}{n}\sum_{j=D(t)}^{t-1}\mathbb{E}\left[\frac{1}{n}\sum_{i\in S}f_i(x^j) - f_i(x^*) - \langle\nabla f_i(x^*), x^j - x^*\rangle\right]$$

$$+ \frac{4L\tau}{n}\mathbb{E}\left[\frac{1}{n}\sum_{i\in S}f_i(\alpha_i^{D(t)}) - f_i(x^*) - \langle\nabla f_i(x^*), \alpha_i^{D(t)} - x^*\rangle\right].$$

The first inequality follows from the fact that if $\alpha_{i_t}^{D(t)}$ and $\alpha_{i_t}^t$ differ, then (a) $i_t$ should have been chosen in one of the iteration $j \in \{D(t), \ldots, t-1\}$ and (b) $\alpha_{i_t}$ is changed to $x^j$ in that iteration. The second inequality follows from Lemma 3 for r = 2. The third inequality follows from the fact that the probability $P(i_j = i) = 1/n$. The last step directly follows from Lemma 1. Note that sum is over indices in $S$ since $\alpha_i$'s for $i \notin S$ do not change during the epoch.

The third term in Equation (A.15) can be bounded by exactly the same technique we used for the second term. The bound, in fact, turns out to identical to second term since $i_t$ is chosen uniformly

random. Combining all the terms we have

$$T_6 \leq 2\mathbb{E}[\|v^t\|^2] + 6L^2\Delta\eta^2\tau \sum_{d=D(t)}^{t-1} \mathbb{E}\left[\|u^d\|^2\right] + \frac{48L}{n} \sum_{j=D(t)}^{t-1} \mathbb{E}\left[D_g(x^j, x^*)\right] + \frac{48L\tau}{n}\mathbb{E}\left[G_{D(t)}\right].$$

The term $T_7$ can be bounded in a manner similar to one in Theorem 2 to obtain the following (see proof of Theorem 2 for details):

$$\mathbb{E}\langle x^t - x^*, u^t \rangle \leq \mathbb{E}\left[f(x^*) - f(x^t) + \frac{L\Delta\tau\eta^2}{2} \sum_{d=D(t)}^{t-1} \|u^d\|^2\right]. \tag{A.16}$$

We need the following bound for our analysis:

$$\sum_{j=0}^{m-1} \left(1 - \frac{1}{\kappa}\right)^{m-1-j} \mathbb{E}[\|u^{km+j}\|^2] \leq 2\sum_{j=0}^{m-1} \left(1 - \frac{1}{\kappa}\right)^{m-1-j} \mathbb{E}[\|v^{km+j}\|^2]$$

$$+ \sum_{t=km}^{km+m-1} 6L^2\Delta\eta^2\tau \sum_{d=D(t)}^{t-1} \mathbb{E}\left[\|u^d\|^2\right]$$

$$+ \sum_{t=km}^{km+m-1} \frac{48L}{n} \sum_{j=D(t)}^{t-1} \mathbb{E}\left[D_g(x^j, x^*)\right]$$

$$+ \sum_{t=km}^{km+m-1} \frac{48L\tau}{n}\mathbb{E}\left[G_{D(t)}\right].$$

The above inequality follows directly from the bound on $T_6$ by adding over all $t$ in the epoch. Under the condition

$$\eta^2 \leq \left(1 - \frac{1}{\kappa}\right)^{m-1} \frac{1}{12L^2\Delta\tau^2}.$$

we have the following inequality

$$\sum_{j=0}^{m-1} \left(1 - \frac{1}{\kappa}\right)^{m-1-j} \mathbb{E}[\|u^{km+j}\|^2] \leq 4\sum_{j=0}^{m-1} \left(1 - \frac{1}{\kappa}\right)^{m-1-j} \mathbb{E}[\|v^{km+j}\|^2]$$

$$+ \sum_{t=km}^{km+m-1} \frac{96L}{n} \sum_{j=D(t)}^{t-1} \mathbb{E}\left[D_g(x^j, x^*)\right]$$

$$+ \sum_{t=km}^{km+m-1} \frac{96L\tau}{n}\mathbb{E}\left[G_{D(t)}\right]. \tag{A.17}$$

The above inequality follows from the fact that

$$\sum_{t=km}^{km+m-1} 6L^2\Delta\eta^2\tau \sum_{d=D(t)}^{t-1} \mathbb{E}\left[\|u^d\|^2\right] \leq \sum_{t=km}^{km+m-1} 6L^2\Delta\eta^2\tau^2\mathbb{E}\left[\|u^t\|^2\right]$$

$$\leq \frac{1}{2}\sum_{j=0}^{m-1} \left(1 - \frac{1}{\kappa}\right)^{m-1-j} \mathbb{E}[\|u^{km+j}\|^2].$$

The above relationship is due to the condition on $\eta$ and the fact that any $d \in \{D(t), \ldots, t-1\}$ for at most $\tau$ values of $t$. We have the following:

$$R_{t+1} = c\mathbb{E}\left[\|x^t - x^*\|^2\right] + c\eta^2\mathbb{E}\left[\|u^t\|^2\right] + 2c\eta\mathbb{E}\left[\langle x^t - x^*, u^t \rangle\right] + \mathbb{E}\left[G_{t+1}\right]$$

$$:= \left(1 - \frac{1}{\kappa}\right)R_t + e_t. \tag{A.18}$$

We bound $e_t$ in the following manner:

$$e_t = \frac{c}{\kappa} \|x^t - x^*\|^2 + \left(\frac{1}{\kappa} - \frac{1}{n}\right) \mathbb{E}[G_t] + c\eta^2 \mathbb{E}\left[\|u^t\|^2\right] + 2c\eta \mathbb{E}\left[\langle x^t - x^*, u^t\rangle\right] + \mathbb{E}[G_{t+1}]$$

$$= \frac{c}{\kappa} \|x^t - x^*\|^2 + \left(\frac{1}{\kappa} - \frac{1}{n}\right) \mathbb{E}[G_t] + c\eta^2 \mathbb{E}\left[\|u^t\|^2\right] + 2c\eta \mathbb{E}\left[\langle x^t - x^*, u^t\rangle\right] + \frac{1}{n}\mathbb{E}[D_g(x^t, x^*)]$$

$$\leq -\left(2c\eta - \frac{2c}{\kappa\lambda}\right) \mathbb{E}\left[f(x^t) - f(x^*)\right] + \left(\frac{1}{\kappa} - \frac{1}{n}\right) \mathbb{E}[G_t] + c\eta^2 \mathbb{E}[\|u^t\|^2]$$

$$+ cL\Delta\tau\eta^3 \sum_{d=D(t)}^{t-1} \mathbb{E}\left[\|u^d\|^2\right] + \frac{1}{n}\mathbb{E}[D_g(x^t, x^*)].$$

The second equality follows from the definition of $G_{t+1}$ (see Equation (A.2)).

$$\mathbb{E}[G_{t+1}] = \left(1 - \frac{1}{n}\right) \mathbb{E}[G_t] + \frac{1}{n}\mathbb{E}[D_g(x^t, x^*)].$$

Applying the recurrence relationship in Equation (A.18) with the derived bound on $e_t$, we have

$$R_{km+m} \leq \left(1 - \frac{1}{\kappa}\right)^m \tilde{R}_k + \sum_{j=0}^{m-1} \left(1 - \frac{1}{\kappa}\right)^{m-1-j} e_{km+j}$$

$$\leq \left(1 - \frac{1}{\kappa}\right)^m \tilde{R}_k + \sum_{j=0}^{m-1} \left(1 - \frac{1}{\kappa}\right)^{m-1-j} e'_{km+j},$$

where $e'_t$ is defined as follows

$$\tilde{R}_k = \mathbb{E}\left[c\|\tilde{x}^k - x^*\|^2 + \tilde{G}_k\right]$$

$$e'_t = -\left(2c\eta - \frac{2c}{\kappa\lambda}\right) \mathbb{E}\left[f(x^t) - f(x^*)\right] + \left(\frac{1}{\kappa} - \frac{1}{n}\right) \mathbb{E}[G_t]$$

$$+ \left(c\eta^2 + \left(1 - \frac{1}{\kappa}\right)^{-\tau} cL\Delta\tau^2\eta^3\right) \mathbb{E}[\|u^t\|^2] + \frac{1}{n}\mathbb{E}[D_g(x^t, x^*)].$$

The last inequality follows from that fact that the delay is at most $\tau$. In particular, each index $j \in \{D(t), \dots, t-1\}$ occurs at most $\tau$ times. We use the following notation for ease of exposition:

$$\zeta = \left(c\eta^2 + \left(1 - \frac{1}{\kappa}\right)^{-\tau} cL\Delta\tau^2\eta^3\right).$$

Substituting the bound in Equation (A.17), we get the following:

$$R_{km+m} \leq \left(1 - \frac{1}{\kappa}\right)^m \tilde{R}_k - \left(2c\eta - \frac{2c}{\kappa\lambda}\right) \sum_{j=0}^{m-1} \left(1 - \frac{1}{\kappa}\right)^{m-1-j} \mathbb{E}\left[f(x^{km+j}) - f(x^*)\right]$$

$$+ 4\zeta \sum_{j=0}^{m-1} \left(1 - \frac{1}{\kappa}\right)^{m-1-j} \mathbb{E}[\|v^{km+j}\|^2]$$

$$+ \left[\frac{96\zeta L\tau}{n}\left(1 - \frac{1}{\kappa}\right)^{-\tau} + \frac{1}{n}\right] \sum_{j=0}^{m-1} \left(1 - \frac{1}{\kappa}\right)^{m-1-j} \mathbb{E}\left[D_g(x^{km+j}, x^*)\right]$$

$$+ \left[\frac{1}{\kappa} + \frac{96\zeta L\tau}{n}\left(1 - \frac{1}{\kappa}\right)^{-\tau} - \frac{1}{n}\right] \sum_{j=0}^{m-1} \left(1 - \frac{1}{\kappa}\right)^{m-1-j} \mathbb{E}\left[G_{D(km+j)}\right]. \quad (A.19)$$

We now use the following previously used bound on $v^t$ (see bound $T_2$ in the proof of Theorem 1):

$$\mathbb{E}[\|v^t\|^2] \leq 2L\left(1 + \frac{1}{\beta}\right)\left[G_t + D_h(\tilde{x}^k, x^*)\right] + 2L(1 + \beta)\mathbb{E}[f(x^t) - f(x^*)].$$

Substituting the above bound on $v^t$ in Equation (A.19), we get the following:

$$
\begin{aligned}
R_{km+m} \leq &\left(1 - \frac{1}{\kappa}\right)^m \tilde{R}_k \\
&- \left[2c\eta - 8\zeta L(1+\beta) - \frac{2c}{\kappa\lambda} - \frac{96\zeta L\tau}{n}\left(1 - \frac{1}{\kappa}\right)^{-\tau} - \frac{1}{n}\right] \times \\
&\qquad \sum_{j=0}^{m-1}\left(1 - \frac{1}{\kappa}\right)^{m-1-j} \mathbb{E}\left[f(x^{km+j}) - f(x^*)\right] \\
&+ \left[\frac{1}{\kappa} + 8\zeta L\left(1 + \frac{1}{\beta}\right) + \frac{96\zeta L\tau}{n}\left(1 - \frac{1}{\kappa}\right)^{-\tau} - \frac{1}{n}\right] \times \\
&\qquad \sum_{j=0}^{m-1}\left(1 - \frac{1}{\kappa}\right)^{m-1-j} \mathbb{E}\left[G_{km+j}\right] \\
&+ 8\zeta L\left(1 + \frac{1}{\beta}\right)\sum_{j=0}^{m-1}\left(1 - \frac{1}{\kappa}\right)^{m-1-j} \mathbb{E}\left[D_h(\tilde{x}^k, x^*)\right] \\
\leq &\frac{2c}{\lambda}\left(1 - \frac{1}{\kappa}\right)^m \mathbb{E}\left[f(\tilde{x}^k) - f(x^*)\right] + \left(1 - \frac{1}{\kappa}\right)^m \mathbb{E}\left[\tilde{G}_k\right] \\
&- \left[2c\eta - 8\zeta L(1+\beta) - \frac{2c}{\kappa\lambda} - \frac{96\zeta L\tau}{n}\left(1 - \frac{1}{\kappa}\right)^{-\tau} - \frac{1}{n}\right] \times \\
&\qquad \sum_{j=0}^{m-1}\left(1 - \frac{1}{\kappa}\right)^{m-1-j} \mathbb{E}\left[f(x^{km+j}) - f(x^*)\right] \\
&+ 8\zeta L\left(1 + \frac{1}{\beta}\right)\kappa\left[1 - \left(1 - \frac{1}{\kappa}\right)^m\right]\mathbb{E}\left[D_h(\tilde{x}^k, x^*)\right]. \qquad \text{(A.20)}
\end{aligned}
$$

The first step is due to the Bregman divergence based inequality $D_f(x, x^*) \geq D_g(x, x^*)$. The second step follows from the expanding $\tilde{R}_k$ and using the strong convexity of function $f$. For brevity, we use the following notation:

$$
\gamma_a = \kappa\left[1 - \left(1 - \frac{1}{\kappa}\right)^m\right]\left[2c\eta - 8\zeta L(1+\beta) - \frac{2c}{\kappa\lambda} - \frac{96\zeta L\tau}{n}\left(1 - \frac{1}{\kappa}\right)^{-\tau} - \frac{1}{n}\right]
$$

$$
\theta_a = \max\left\{\left[\frac{2c}{\gamma_a\lambda}\left(1 - \frac{1}{\kappa}\right)^m + \frac{8\zeta L\left(1 + \frac{1}{\beta}\right)}{\gamma_a}\kappa\left[1 - \left(1 - \frac{1}{\kappa}\right)^m\right]\right], \left(1 - \frac{1}{\kappa}\right)^m\right\}.
$$

We now use the fact that $\tilde{x}^{k+1}$ is chosen randomly from $\{x^{km}, \ldots, x^{km+m-1}\}$ with probabilities proportional to $\{(1 - 1/\kappa)^{m-1}, \ldots, 1\}$. Hence, we have the following inequality from Equation (A.20):

$$
\mathbb{E}\left[f(\tilde{x}^{k+1}) - f(x^*) + \frac{1}{\gamma_a}\tilde{G}_{k+1}\right] \leq \theta_a \mathbb{E}\left[f(\tilde{x}^k) - f(x^*) + \frac{1}{\gamma_a}\tilde{G}_k\right],
$$

where $\theta_a < 1$ is a constant that depends on the parameters used in the algorithm. $\qquad\square$

**Remarks about the parameters in Theorem 1 & Theorem 3**

In this section, we briefly remark about the parameters in Theorems 1 & 3. For Theorem 1, suppose we use the following instantiation of the parameters:

$$\eta = \frac{1}{16(\lambda n + L)}$$

$$\kappa = \frac{4}{\lambda \eta} = 64\left(n + \frac{L}{\lambda}\right)$$

$$\beta = \frac{2\lambda n + L}{L}$$

$$c = \frac{2}{\eta n} = 32\left(\lambda + \frac{L}{n}\right).$$

Then we have,

$$\theta = \max\left\{\left[\frac{2(1-\frac{1}{\kappa})^m}{3\left(1-(1-\frac{1}{\kappa})^m\right)} + \frac{1}{3\left(1+\frac{2\lambda n}{L}\right)}\right], \left(1-\frac{1}{\kappa}\right)^m\right\}.$$

In the interesting case of $L/\lambda = n$ (high condition number regime), since $\kappa = \Theta(n)$, one can obtain a constant $\theta$ (say $\theta = 0.5$) with $m = O(n)$. This leads to $\epsilon$ accuracy in the objective function after $O(\log(1/\epsilon))$ epochs of HSAG. When $m = O(n)$, the computational complexity of each epoch of HSAG is $O(n)$. Hence, the total computational complexity of HSAG is $O(n\log(1/\epsilon))$. On the other hand, because $L/\lambda = n$, batch gradient descent method requires $O(n\log(1/\epsilon))$ iterations to achieve $\epsilon$ accuracy in the objective value. Since the complexity of each iteration of gradient descent is $O(n)$ (as it passes through the whole dataset for calculating the gradient), the overall computational complexity of batch gradient descent is $O(n^2\log(1/\epsilon))$. In general, for high condition number regimes (which is typically the case in machine learning applications), HSAG (like SVRG, SAGA) will be significantly faster than the batch gradient methods. Furthermore, the convergence rate is strictly better than the sublinear rate obtained for SGD.

The parameter instantiations for Theorem 3 are much more involved. Suppose $\Delta^{1/2}\tau < 1$ (this is the sparse regime that is typically of interest to the machine learning community) and $m > n > 9\tau$. The other case ($\Delta^{1/2}\tau \geq 1$) can be analyzed in a similar fashion. We set the following parameters:

$$\eta = \frac{\left(1-\frac{1}{\kappa}\right)^m}{64(\lambda n + L)}$$

$$\kappa = \frac{4}{\lambda \eta} = \frac{256\left(n + \frac{L}{\lambda}\right)}{(1-\frac{1}{\kappa})^m}$$

$$\beta = \frac{2\lambda n + L}{L}$$

$$c = \frac{2}{\eta n} = 32\left(\lambda + \frac{L}{n}\right).$$

Then we have the following:

$$\zeta \leq \frac{\left(1-\frac{1}{\kappa}\right)^m}{32n(\lambda n + L)}\left(1 + \frac{L}{64(\lambda n + L)}\right)$$

$$\theta_a \leq \max\left\{\left[\frac{6(1-\frac{1}{\kappa})^m}{7\left(1-(1-\frac{1}{\kappa})^m\right)} + \frac{195\left(1-\frac{1}{\kappa}\right)^\tau}{448(1+\frac{2\lambda n}{L})}\right], \left(1-\frac{1}{\kappa}\right)^m\right\}$$

Again, in the case of $L/\lambda = n$, we can obtain constant $\theta_a$ (say $\theta_a = 0.5$) with $m = \Theta(n)$ and $\kappa = \Theta(n)$. The constants in the parameters are not optimized and can be improved by a more careful analysis. Furthermore, sharper constants can be obtained in specific cases. For example, see [10] and Theorem 2 for synchronous and asynchronous convergence rates of SVRG respectively. Similarly, sharper constants for SAGA can also be derived by simple modifications of the analysis.

## Other Lemmatta

**Lemma 1.** *[10] For any $\alpha_i \in \mathbb{R}^d$ where $i \in [n]$ and $x^*$, we have*

$$\mathbb{E}\left[\|\nabla f_{i_t}(\alpha_{i_t}) - \nabla f_{i_t}(x^*)\|^2\right] \leq \frac{2L}{n} \sum_i \left[f_i(\alpha_i) - f(x^*) - \langle \nabla f_i(x^*), \alpha_i - x^* \rangle\right].$$

**Lemma 2.** *Suppose $f : \mathbb{R}^d \to \mathbb{R}$ and $f = g + h$ where $f, g$ and $h$ are convex and differentiable. $x^*$ is the optimal solution to $\arg\min_x f(x)$ then we have the following*

$$D_f(x, x^*) = f(x) - f(x^*)$$
$$D_f(x, x^*) = D_g(x, x^*) + D_h(x, x^*)$$
$$D_f(x, x^*) \geq D_g(x, x^*).$$

*Proof.* The proof follows trivially from the fact that $x^*$ is the optimal solution and linearity and non-negative properties of Bregman divergence. $\qquad\square$

**Lemma 3.** *For random variables $z_1, \ldots, z_r$, we have*

$$\mathbb{E}\left[\|z_1 + \ldots + z_r\|^2\right] \leq r\mathbb{E}\left[\|z_1\|^2 + \ldots + \|z_r\|^2\right].$$