[Reviews · NeurIPS 2015]

Submitted by Assigned_Reviewer_1

"Light" review:

This work proposes a framework for linearly-convergent stochastic gradient methods that allows asynchronous computation. The paper analyzes a generalization of SVRG and SAGA, showing what speedups can be obtained.

I like the statement of the general framework, and how it includes many of the previous algorithms. However, I think if you made two small modifications you could include basically all of the linearly-convergent primal methods:

1. The framework doesn't seem to include SAG because the update is a bit different. In particular, SAG weighs the two $\nabla f_{i_t}$ terms by (1/n). If you added a weight to Algorithm 1 on these terms, it would allow you to include SAG. Further, adding these weight would allow you to include non-uniform sampling in the framework as in [27].

2. In the update x^{t+1} = x^t - \eta(...), you could replace x^t by a weighted average of x^t and the alpha_i^t. If you do this, you could include MISO in the framework. MISO would correspond to weights of 1/n on the alpha_i^t, and all other methods place all their weight on x^t.

I did not check the details, but all the results seems reasonable and plausible.

The experiments are not terribly convincing in terms of utility of the method. On most datasets there only seems to be a small improvement over the locked SVRG (less than a factor of 2 in 3/4 cases). It would be interesting to see the difference if a larger number of threads were available.
Summary: The problem is important, and the proposed framework seems useful to think about these methods. The theoretical results seem correct although not surprising, and the experimental results show a small advantage of the proposed asynchronous strategy.

Submitted by Assigned_Reviewer_2

This paper first proposes a general analysis of variance reducing stochastic gradient techniques. This leads to the introduction of a hybrid algorithm, HSAG, then of its asynchronous parallel variant.

The idea of providing a general framework for these methods and their distribution is appealing. Further, the practical impact of a distributed version is also large. However, this work suffers from several weaknesses which makes me vote for rejection: - A common analysis of all these methods has already been provided in the SAGA paper, which you cite, in a clearer way. Due to the way you describe the algorithms, it is hard to understand what the true differences are. - The motivation for HSAG is difficult to understand. You introduce it to combine the appeals of SAGA and SVRG. However, in practice, if memory is an issue, the set S will likely be very small and there will be no significant difference with SVRG. - Section 3 is clearly the most interesting of the paper. Unfortunately, it is really hard to get a grasp of what is going on. Save for one paragraph, no intuition is given on the constants appearing in the theorems. It would have been interesting, for instance, to compare the bounds obtained in theorems 2 and 3.

Finally, with a proper distribution scheme, it is likely that the memory cost will be less of an issue since it will be split across nodes.
Summary: This paper adresses the distribution of variance-reducing methods, which is of clear interest to the optimization community.

Submitted by Assigned_Reviewer_3

Summary: The authors propose a general framework for variance reduction stochastic methods, including SVRG, SAGA, SAG and gradient descent. They also demonstrate the convergence results for hybrid stochastic average gradient, a special case of the proposed framework. In addition, asynchronous versions of stochastic methods are also studied in the paper.

The paper is well presented and easy to follow. The general description of various stochastic method is novel and the convergence results on HSAG is interesting. Asynchronous variance reduction algorithms are also important complements to the stochastic learning research.

I have some concerns about the significance of the results: (1) although the general framework is mathematically interesting, it does not provide much insight to the stochastic learning problem itself. Will this framework help design algorithms that gain better performance in any specific scenario? (2) the value of the asynchronous algorithms are not well justified. The performance of this single machine parallel algorithm, when data scales up, is not properly studied. How it compares with recent work on distributed stochastic algorithms? The data sets used in the experiments are too small to show the practical value.
Summary: This paper shows interesting connections between multiple stochastic gradient algorithms and designs a unified algorithm. In addition, asynchronous versions of the proposed framework and a special case are investigated. The results are nice. However, the impact of the results in the current version is limited.

Author Feedback
Author rebuttal: We thank all the reviewers for their comments. We will gladly address their concerns + suggestions in the final version.

*SUMMARY*
But first let us summarize the crucial points of our response.

1. There is consensus among most reviewers that the paper solves an important problem.
2. We make important advances in incremental variance reduced (VR) methods. We make a core conceptual contribution by analyzing VR methods under a common umbrella (something missing in previous works). We provide the first analysis for various VR methods in an asynchronous setting. Analysis of asynchronous VR algorithms has been noted as an important open problem (see Defazio's thesis [4] and references therein). We give a positive answer to this question and provide exact characterization of the rates.
3. Experimental results: Our experimental focus was to emphasize that benefits of VR methods over SGD also extend to asynchronous settings. In general, SVRG and SAGA have been observed to perform fairly similarly; we expect their asynchronous setting to be similar. We are happy to provide more comparisons between various VR methods in the final version if this a point of contention.

*REV_1*

We are sorry about the slight confusion in the definition of x^{km+m}. As mentioned in line 180-181, the analysis is for the case where the iterate at the end of an epoch is replaced with an average of the iterates in the epoch. The seminal paper [9] has a similar difference between the algorithm and its analysis (see Options 1, 2 in Fig.1 of [9]). We will edit the text to make this point clearer.

Section 3 outlines the steps of our asynchronous algorithm. Each processor runs the 4 steps mentioned in lines 224-229 concurrently. The subroutines for updating x^t and A are shown in Alg.1 and Figs 1 & 2, respectively. We make only fairly standard assumptions of consistent read and bounded delay (please see [14,20]). No other assumptions are made in the paper.

Condition numbers depending on the number of samples are fairly common in ML. Typically, the regularization constant lambda is decreased with n (this can also be seen in learning theory bounds). References [9,5] also discuss details about analysis in this setting. While we focus on the high condition number regime, our theoretical results are good in other regimes as well.

Regarding experiments, please refer to point 3 in the summary.

*REV_2*

There are few serious misunderstandings in the review. We would like to clarify these. First, we stress that a general analysis for VR algorithms is not provided in the SAGA paper. It only discusses a general view of VR algorithms, but it neither develops a formal framework nor analyzes algorithms in this framework.

There are crucial differences with Mahajan et al [MAKSB]:

(a) Our focus is somewhat different. Our analysis is for the asynchronous parallel setting, while MAKSB mainly consider distributed settings (with focus on tradeoffs between communication and computation). Moreover, their algorithm is *not* asynchronous.

(b) Our theoretical results are stronger. In particular, the convergence result in Eqn (30) of MAKSB (http://arxiv.org/abs/1310.8418) is in spirit, only as strong as Gradient Descent (GD). But VR methods (including ours) perform better than GD by an order of magnitude (typically order n; please refer to the discussion after proof of Theorem 1 in [9] and lines 211-215 of our paper).

Lines 211-215 and 262-275 gives explicit values of the constants and the corresponding theoretical results. Our results are similar to SVRG, SAGA for the synchronous case. We will add more details about the constants and a comparison of Theorem 2 & 3 to the final version.

We are happy to include more experimental comparisons too. Please refer to point 2 in the summary.

*REV_3*

First, we'd like to argue that the finite sum problem is interesting in its own right. In addition to capturing many ML problems, it is also observed that the test error of VR methods (proxy for generalization error) is often better than SGD (please refer to [5,9] for more details). Second, there has been some exciting recent progress on application of VR methods to the stochastic setting (see eg. http://arxiv.org/abs/1412.6606). In light of these advances, we believe that our algorithms have implications in the stochastic learning setting as well.

The implications for ERM (the default problem in statistical ML) are important too. VR methods, along with better empirical performance, greatly ease the problem of designing learning rates --- a painstaking process for SGD and its asynchronous variants (especially for large-scale data). We will add more discussion about this to the paper.

Regarding experiments, please refer to point 2 in the summary.

*REV_4*

We politely disagree with the reviewer, and answer the concerns as a part of points 1 & 2 in the summary above.